# Genome Instability Induced by Topoisomerase Misfunction

**DOI:** 10.3390/ijms251910247

**Published:** 2024-09-24

**Authors:** Karin C. Nitiss, Afif Bandak, James M. Berger, John L. Nitiss

**Affiliations:** 1Pharmaceutical Sciences Department, University of Illinois Chicago, Rockford, IL 61107, USA; knitiss@uic.edu; 2Department of Biophysics and Biophysical Chemistry, Johns Hopkins School of Medicine, Baltimore, MD 20215, USA; afif.bandak@gmail.com (A.B.); jmberger@jhmi.edu (J.M.B.)

**Keywords:** topoisomerase, mutation, genome instability, Top1, Top2

## Abstract

Topoisomerases alter DNA topology by making transient DNA strand breaks (DSBs) in DNA. The DNA cleavage reaction mechanism includes the formation of a reversible protein/DNA complex that allows rapid resealing of the transient break. This mechanism allows changes in DNA topology with minimal risks of persistent DNA damage. Nonetheless, small molecules, alternate DNA structures, or mutations in topoisomerase proteins can impede the resealing of the transient breaks, leading to genome instability and potentially cell death. The consequences of high levels of enzyme/DNA adducts differ for type I and type II topoisomerases. Top1 action on DNA containing ribonucleotides leads to 2–5 nucleotide deletions in repeated sequences, while mutant Top1 enzymes can generate large deletions. By contrast, small molecules that target Top2, or mutant Top2 enzymes with elevated levels of cleavage lead to small *de* novo duplications. Both Top1 and Top2 have the potential to generate large rearrangements and translocations. Thus, genome instability due to topoisomerase mis-function is a potential pathogenic mechanism especially leading to oncogenic progression. Recent studies support the potential roles of topoisomerases in genetic changes in cancer cells, highlighting the need to understand how cells limit genome instability induced by topoisomerases. This review highlights recent studies that bear on these questions.

## 1. Introduction

Metabolic events with double-stranded DNA pose unique challenges since these processes require substantial DNA unwinding. During transcription or replication, enzymes tracking along DNA lead to overwinding of the DNA ahead of the tracking proteins [1,2]. DNA topoisomerases introduce breaks in DNA to alter the topological structures introduced by events such as transcription and replication that might otherwise impede further DNA metabolism [3,4,5]. Topoisomerases break DNA by forming a transient covalent protein/DNA intermediate [6]. The transient break allows strand passage that leads to restoration of normal DNA winding and importantly, allows breaks to be made without provoking DNA damage responses or the generation of genome rearrangements [7,8]. Changes in DNA topology are mediated by two classes of topoisomerases, type I enzymes that generate single-strand breaks and type II enzymes that generate double-strand breaks. Human cells encode two type II topoisomerases termed Top2α and Top2β [3,9,10]. These enzymes have distinct biological functions but share significant sequence homology and act by the same biochemical mechanisms. Normal topoisomerase function is critical for genome stability and has been extensively discussed in recent reviews [8,11,12]. This review discusses current perspectives on genome instability induced by type I and type II enzymes, with an emphasis on aberrant function.

While the topoisomerase mechanism largely avoids generating DNA damage, topoisomerases can damage DNA should they fail to religate the DNA break. The consequences of inappropriately long-lived topoisomerase breaks can be seen when cells are treated with anti-cancer drugs such as topotecan that inhibits DNA religation by type I topoisomerases or etoposide that inhibits DNA religation by type II topoisomerases [13]. The trapped DNA/topoisomerase covalent complex (cleavage complex) can block transcription and replication, induce DNA damage signals, induce senescence and apoptosis, and if misrepaired, can lead to genome instability (reviewed in [13,14,15]). This mechanism is the basis for the action of anti-cancer agents that target eukaryotic topoisomerases and fluoroquinolone antibiotics that act against bacterial enzymes. Because agents such as camptothecins and topoisomerase II targeting agents produce protein/DNA lesions, these agents have been termed topoisomerase poisons. This mechanism is in contrast to small molecules that act as enzyme inhibitors without elevating the levels of protein/DNA complexes [16]. The structural studies of both Top1 and Top2 poisons indicate that small molecule inhibition frequently occurs at the interface of DNA and protein [17], and prevents the enzyme from returning to an intermediate where DNA ligation can occur [18,19]. Since the overall reaction mechanism for topoisomerases includes an orchestrated series of steps that involves substantial movement of the protein (and DNA substrates), it is possible that ligation could be inhibited by action at some distance from the DNA. This is supported by observations that mutations that affect the action of Top2 poisons frequently occur far from the protein/DNA interface [20].

In addition to small molecule inhibitors, a variety of other mechanisms can lead to elevated levels of trapped topoisomerases. These include damage to DNA bases or DNA breaks [7]. Misincorporation of ribonucleotides has been shown to trap both type I and type II topoisomerases [21,22]. Recent studies also suggest that different DNA structures such as G4 quadruplex DNA may also affect topoisomerase-mediated DNA cleavage [23].

Work from our laboratories has identified mutations in eukaryotic topoisomerases that lead to elevated cleavage independent of small molecules [24,25]. Similar mutations previously described in eukaryotic topoisomerase I [26] have proven invaluable for dissecting the consequences of topoisomerases that generate high levels of DNA cleavage. Topoisomerases with high levels of DNA cleavage may also have physiological relevance. Notably, it has recently been suggested that Top2β generates long-lasting breaks at promoters [27,28,29,30]. Therefore, DNA damage induced by topoisomerases in the absence of inhibitors may be an important aspect of topoisomerase biology and biochemistry [7].

Since topoisomerases can generate DNA damage, either by small molecule inhibitors or by other mechanisms that lead to elevated cleavage, it is plausible that topoisomerases could be oncogenic drivers. Until recently, the strongest evidence has been secondary malignancies induced by Top2 targeting drugs such as etoposide and mitoxantrone [31,32,33]. 

In this review, we broadly consider the ways that topoisomerase mis-functioning can impact genome stability. We describe genome instability induced by Top1 mis-function, especially the induction of deletions by Top1 action at sites of ribonucleotide mis-incorporation into DNA. Recent studies from our laboratories that highlight the effects of mutant Top2 are discussed. We summarize what we have learned using hyper cleavage topoisomerase mutants and consider the evidence that alterations in either Top1 or Top2 can lead to pathological consequences. We also consider how the effects of these mutants may illuminate biological consequences when topoisomerases generate high levels of DNA cleavage.

## 2. Top1 and Genome Instability

Topoisomerase function is required for most DNA transactions, so it is not surprising that loss-of-function mutations in topoisomerase genes results in genome instability. Neither *Saccharomyces cerevisiae* nor *Schizosaccharomyces pombe* requires *Top1* for viability [34,35], while Top1 is essential, at least in proliferating cells, in higher eukaryotes [8]. While detailed studies have been most extensive in yeast systems, siRNA knockdown experiments have shown the importance of Top1 in genome stability in higher eukaryotes. For example, R-loops (RNA–DNA hybrids with a single-strand loop) form at elevated levels when Top1 is depleted [36]. For both yeast and higher eukaryotes, elevated levels of DNA supercoiling or the formation of non-B-DNA structures would promote genome instability [37,38,39]. Yeast cells lacking Top1 have elevated levels of recombination, but only in genes encoding ribosomal RNAs [40]. Genes encoding ribosomal DNA have unique properties; they are highly expressed and present in yeast as a single large tandem array. Specific sequences block bidirectional replication within the cluster, minimizing transcription: replication conflicts [41]. Interestingly, a unique reaction occurs in cells lacking Top1 and with reduced levels of Top2. The repeat units of ribosomal DNA can be excised as unit-length circles that are maintained episomally. While wild-type yeast cells have low levels of episomal ribosomal DNA circles, the levels are greatly elevated in *top1 top2* double mutants [42]. It has been suggested that elevated levels of episomal ribosomal circles lead to more rapid aging in yeast cells [43]. The importance of topoisomerase-induced generation of such episomes in higher organisms requires additional study [44]

The study of genome instability induced by topoisomerases was greatly assisted by the demonstration that small molecules could trap topoisomerases as cleavage complexes and that cells treated with these agents exhibited genome instability [45,46]. As noted in the introduction, such small molecules, such as camptothecins, are referred to as Top1 poisons. For Top1 targeting agents, an important model for the induction of genome instability is the induction of double-strand breaks when a replication fork collides with a trapped Top1 complex [47]. Unlike the loss of function mutants of topoisomerase I, the treatment of yeast cells with camptothecin results in elevated genome instability at all sites in the genome [46]. In yeast, this largely manifested as induction of homologous recombination and generation of small deletions. However, there are other ways that camptothecin-trapped Top1 can lead to genome instability (reviewed in [11]). Recent work has highlighted that replication-independent mechanisms of Top1-induced genome instability may be induced by transcription-mediated DNA damage (reviewed in [48]).

In addition to small molecule inhibitors, structural alterations in DNA can result in trapping of Top1 on DNA [14]. These structural alterations can include a wide range of lesions in DNA such as pyrimidine dimers, alkylated bases and abasic sites. Top1 acting at these sites of damage has been shown to lead to genomic alterations [49]. 

A landmark in the study of Top1-induced genome instability was the finding that Top1 was responsible for the induction of mutations in highly transcribed genes in yeast [50,51]. The mutational signature of these Top1 dependent events consisted of 2–5 bp deletions at tandem repeats where the deletion size matched the repeat unit. A major pathway of Top1-induced 2–5 bp deletions was shown to arise from ribonucleotide mis-incorporation in DNA [52]. Replicative polymerases will occasionally incorporate ribonucleotides instead of deoxyribonucleotides into DNA, which is expected given the much higher levels of ribonucleotides in cells [53]. Previous work had demonstrated that type 1B topoisomerases could cleave DNA containing a ribonucleotide and that Top1 could then be removed by a reaction with the 2′ OH of the ribose sugar to yield a free 2′, 3′ cyclic phosphate product, along with the release of the topoisomerase protein [54]. This blocked lesion can be acted upon by a second Top1 cleavage reaction (Figure 1). The cleavage 5’ of the 2′, 3′ cyclic phosphate with its subsequent loss, followed by Top1-mediated religation results in the removal of one repeat unit. Therefore, ribonucleotides are a unique lesion in DNA with respect to topoisomerase cleavage since the 2′ OH of the ribose sugar provokes a mechanism of removal of the protein covalently bound to the nucleic acid. This mechanism has been proposed as an alternative (error-prone) pathway for the removal of ribonucleotides [55,56]. The canonical pathway for removal of mis-incorporated ribonucleotides depends on RNaseH2 in a repair pathway termed ribonucleotide excision repair (reviewed in [22,57]). The mutational signal of Top1-induced events, namely 2–5 bp deletions at tandem repeats, has been applied to demonstrate that these events also occur in human cancers. The significance of these findings is discussed below.

Alterations in the Top1 protein can also lead to enhanced levels of DNA cleavage. Several mutants have been described in yeast and mammalian Top1 that lead to enzymes that are hyper-cleavage in vitro and induce DNA damage in cells. Expression of proteins such as the Top1(T722A) results in elevated levels of recombination throughout the yeast genome and therefore not surprisingly, expression of Top1(T722A) and other hype-cleavage alleles is lethal in yeast strains defective in homologous recombination, such as *rad52^−^* cells [58,59]. Interestingly, other types of genome instability have been found in yeast cells expressing hyper cleavage Top1 alleles. Cho and Jinks-Robertson identified large deletions induced by Top1(T722A). They identified deletions ranging from about 100–500 nucleotides and a second class of events that included deletions of up to about 4.1 kb [60]. These 4.1 kb deletions were driven by the specific detection system used, therefore the upper limit of deletion size by hyper cleavage Top1 enzymes was not determined. The deletion events required NHEJ, and therefore occurred through a double-strand break intermediate [60]. The events had several other noteworthy characteristics, including occurrence in non-dividing cells and a limited dependence on ribonucleotide misincorporation. The authors suggest that the events arise from two independent DNA double-strand breaks induced by the hyper cleavage enzymes; in non-dividing cells, the lack of a sister chromatid eliminates the possibility of repair by homologous recombination. It seems reasonable from these experiments that Top1 cleavage, when it leads to double-strand breaks, has the potential to generate DNA translocations.

Taken together, Top1 can generate genome instability by double-strand break induction, either by small molecule inhibitors or by mutant enzymes. By contrast, the action of Top1 at ribonucleotides leads to genome instability that is largely independent of double-strand breaks. While the standard model of Top1-induced double-strand breaks occurs primarily when a replication fork collides with a trapped Top1 complex, events induced by hyper cleavage Top1 alleles include double-strand breaks that arise independent of replication. Other recent work with camptothecin-induced damage also suggests the induction of double-strand breaks by replication-independent mechanisms [61,62]. While the mechanisms of double-strand break induction in the absence of replication remain unclear, a possible mechanism may be Top1 cleavage in the vicinity of a pre-existing DNA strand break [63]. If Top1 poisons can induce double-strand breaks in non-replicating cells, one might expect that Top1 poisons may also lead to secondary malignancies as do Top2 poisons such as etoposide. However, induction of high levels of secondary malignancies by camptothecins has not yet been reported.

Since Top1 mis-functioning can lead to genome instability, a plausible hypothesis is that Top1-induced mutations might be found in cancer cells thereby accelerating tumorigenesis. Ribonucleotide-induced events give rise to a specific type of mutation, i.e., deletions of 2–5 bp as described above. This pattern is the same as ID4 (Insertion/Deletion 4). Reijns and colleagues have recently described extensive evidence that ID4 arises from Top1 action at ribonucleotides including the demonstration that ID4 arises when ribonucleotides are mis-incorporated into DNA [64]. The mutations found as part of the ID4 signature occur most frequently in highly expressed genes, as was described for the mutations induced by ribonucleotides in yeast. Although the importance of Top1-induced mutations in cancer requires further exploration, it is clear that Top1 is the driver of specific mutational processes, with the mutations induced by Top1 action seen in cancer cells. Chronic lymphocytic leukemias are often associated with loss of RnaseH2, the key enzyme required for normal ribonucleotide excision repair, thus some cancers may have a mutator phenotype due to the action of Top1 removal of ribonucleotides. Further studies will be needed to explore the different contexts where Top1 mutagenesis is important. Finally, hyper cleavage Top1 alleles that are expressed in human cancers have not yet been described.

## 3. Top2 and Genome Stability

Unlike Top1, a Top2 isoform is required for viability for all eukaryotes. Top2 is required for separating replicated chromosomes at mitosis, and is not essential at other points in the cell cycle (reviewed in [4,65]). Therefore, studying the consequences of Top2 deficiency has required either the use of conditional mutants (such as yeast temperature-sensitive mutants) or small molecule inhibitors that do not lead to elevated levels of covalent complexes. The small molecules of choice have been primarily bisdioxopiperazines such as ICRF-193 [66] or merbarone [67], and these agents are termed catalytic inhibitors. Both agents have potential issues including the potential for DNA damage independent of covalent complex formation [68,69] and lack of specificity for targeting Top2. Importantly, the mechanism of action of merbarone, suggested to be due to inhibition of Top2 cleavage [70] has not been explained in molecular detail. 

Studies with yeast temperature-sensitive top2 mutants demonstrated consequences for proliferating cells lacking Top2 activity. As was the case for Top1 deficiency, lack of Top2 activity destabilizes genes encoding ribosomal RNA. Cells lacking Top2 activity can undergo both chromosome mis-segregation and chromosome breakage [71].

Similar to camptothecins, the most clinically active Top2 targeting drugs are topoisomerase poisons. Early studies with Top2 targeting drugs focused on the induction of DNA damage, especially double-strand breaks by agents such as etoposide and anthracyclines [72,73]. Not surprisingly, drugs targeting Top2 were shown in a variety of systems to be potent clastogens [74]. Subsequent studies highlighted the potential for Top2 targeting drugs to induce mutations and lead to unique mutational spectra in treated cells [75,76,77].

The genotoxicity of Top2 targeting agents is a significant clinical issue because all these agents have been shown to induce secondary malignancies (reviewed in [33,78]). The detailed mechanisms of translocation induction by Top2 targeting drugs are beyond the scope of this review, but several salient points are worth highlighting. There is a clear association between sites of cleavage by Top2 in vitro, and sites of translocations found in human cells [79]. This correlation suggests Top2 is a direct actor in the translocations. Metabolites of drugs such as etoposide may be particularly relevant to the induction of potentially oncogenic translocations [80]. This consideration suggests that in vitro results need to be interpreted with some caution until the metabolism of the agents is fully understood. Interestingly, different Top2 targeting agents yield different spectra of translocation [33]. The importance of drug specificity for specific translocations needs further exploration. An appealing hypothesis is that the difference between different drugs may relate to the stability of the covalent complexes. Another speculation suggests that some of the difference may relate to single versus double-strand cleavage by Top2. It remains underappreciated that Top2 in the presence of various inhibitors give rise to both single and double-strand breaks [81]. The details of translocation mechanisms induced by Top2 poisons remain to be explored in detail; however, recent work has suggested that the three-dimensional organization of chromosomes in the nucleus is a major determinant of specific translocation partners [82]. It has also been suggested that certain repair pathways for Top2 damage are error-prone ([83], see Section 6 below for further discussion of this point).

Finally, it has been hypothesized that the Top2β isoform of human Top2 is principally responsible for the induction of secondary malignancies [84], leading to intensive efforts to identify small molecules that specifically target Top2α.

Like Top1, Top2 can also be trapped by structural alterations in DNA. For example, abasic sites greatly stimulate levels of Top2 covalent complexes [85,86]. Interestingly, abasic sites do not appear to inhibit enzyme-mediated re-ligation, but likely enhance overall levels of DNA cleavage. Alternate DNA structures such as G-quadruplex DNA also alter DNA cleavage and re-ligation. Like Top1, the presence of ribonucleotides enhances DNA cleavage by Top2 [87]. The 5′-phosphotyrosyl intermediate has not been reported to form a blocked intermediate, as described above for Top1. Therefore, any trapping of Top2 by ribonucleotides in DNA will require the removal of the protein adduct. Unlike the effects of Top1 trapping by DNA damage, there is less evidence that Top2 enhances cell killing in the presence of DNA damage [88,89,90]. One possibility is that DNA damage does not typically block religation by Top2, perhaps lessening the probability of interference with DNA metabolism [81].

## 4. Top2 and Genome Instability: The Damage Induced by Defective Enzymes

During studies related to the action of catalytic inhibitors of Top2, a mutation in human Top2α, Asp48Asn was identified that could not be viably expressed in recombination-deficient yeast cells [91]. While the purified mutant protein had elevated DNA cleavage in vitro, the orthologous yeast mutant had limited effects and was not studied further. Subsequently, we identified two independent mutants of yeast Top2-R1128G, abbreviated top2-R,G, and separately Top2 (F1025Y R1128G), top2-FY,RG that were also unable to be viably expressed in *rad52^−^* cells. Expression of these mutants in yeast led to elevated levels of recombination. Further, yeast cells expressing top2-FY,RG, had elevated levels of Top2p covalently bound to DNA in cells [24]. These results demonstrated that these mutant proteins caused genome instability, and specifically led to Top2-mediated DNA damage. As with the Top2α, Asp48Asn protein, top2-R,G and top2-FY,RG exhibited elevated levels of drug-independent DNA cleavage [24].

Since the mutant protein generated DNA damage mediated by Top2, we explored the induction of genome instability by top2-FY,RG in collaboration with the Jinks-Robertson laboratory. A simple assay for mutation induction in yeast selects for resistance to the toxic arginine analog canavanine; resistance is conferred mainly by loss-of-function mutations in the CAN1 gene which encodes arginine permease [92]. Careful quantitative analysis of mutation in yeast typically relies on the determination of mutation rates by fluctuation analysis [93]. The obtained mutations can then be sequenced to determine the specific nature of the mutations. The CAN1 forward mutation assay was used to measure mutation rates and analyze mutation types in cells expressing top2-FY,RG. The overall mutation rate was increased approximately 4–5 fold. While cells expressing wild-type Top2 gave rise mainly to base pair substitutions, expression of top2-FY,RG resulted in many insertions of more than one base pair (75/176 mutants sequenced [24]). Notably, most of these insertions (49/75) corresponded to de novo duplications, which are defined as the creation of a repeat where one did not previously exist. Based on the events we observed, we proposed a model for the insertions (Figure 2). The covalently bound Top2 is removed by a nucleolytic function such as Tdp1 (yeast cells lack Tdp2). Our previous work demonstrated that yeast Tdp1 can process Top2 lesions [94]. Removal of the protein results in a double-strand break with a four-nucleotide overhang. The free 3’OH primes DNA synthesis, most likely by a DNA polymerase that functions in DNA repair (e.g., yeast *POL4*). After filling in the four-nucleotide overhang, the blunt ends are ligated by non-homologous end joining (NHEJ). Similar processes can be invoked to explain two and three-nucleotide insertions. This model was tested by examining events in various yeast mutants. For example, cells carrying a deletion of *LIG4*, the DNA ligase that functions in NHEJ, the insertions seen in wild-type cells were not observed. Similarly, cells lacking *POL4* or *TDP1* also exhibited greatly reduced levels of the 2–4 nucleotide insertions. 

The induction of the de novo duplications by top2-FY,RG mimics a process that occurs with wild-type yeast Top2 albeit with greatly enhanced frequency in top2-FY,RG. Overexpression of wild-type Top2 also leads to elevated levels of 2–4 nucleotide insertions as measured by mutations in CAN1. Treatment of yeast cells with the Top2 targeting drug etoposide led to a further increase in insertion events [24]. The sites of the de novo insertions induced in the presence of top2-FY,RG appeared to be non-random, with potential hotspots. The hotspots could arise from a variety of mechanisms including potential preferential sites of Top2 cleavage They could also arise if some sites are preferentially repaired by the error-prone pathway as detailed in Figure 2. 

The generation of a unique mutational signature, namely 2–4 nucleotide insertions, provided an opportunity to examine human cancer sequences for evidence of Top2-induced mutations. An insertion/deletion signature, ID17 shows this pattern [95]. Cancer cells with ID17 were found to carry a mutation in Top2α Lys743Asn (Top2α-K743N) [25]. The validation of Top2α-K743N relied on the yeast system described above for top2-FY,RG. The orthologous mutant of Top2α-K743N is yeast top2-K720N. Expression of top2-K720N is lethal in *rad52^−^* cells. Interestingly, expression of human Top2α-K743N is lethal even in wild-type yeast cells. Expression of yeast top2-K720N gives rise to a strong mutator phenotype. The induction of mutations in cells expressing top2-K720N requires NHEJ. The levels of mutations are reduced in cells lacking POL4 or TDP1. The mutations consist of de novo duplications of 2–4 nucleotides, as described above for top2-FY,RG. Taken together, these results clearly implicate Top2α-K743N as a DNA-damaging topoisomerase that generates mutations that may accelerate oncogenesis [25].

The description above of top2-FY,RG and top2-K720N has left out some important details that require further exploration. top2-FY,RG gives rise to some de novo duplications larger than four nucleotides, mainly five nucleotides. Like the 2–4 nucleotide insertions, the larger de novo duplications require NHEJ [24]. It is unclear how Top2 cleavage can give rise to these larger duplications. Even more intriguing, top2-K720N does not give rise to these larger events. This result suggests that the nature of the mutations depends in part on the damage-prone topoisomerase. Understanding these details will likely unlock additional mechanistic insights. The generation of 2–4 nucleotide insertions is not the only mutational event seen in human cancers expressing Top2α-K743N. The tumors expressing Top2α-K743N also lead to substantial numbers of deletions. The deletions include deletions of length ≥5 bp as well as smaller deletions. These deletions were not found in yeast and qualitatively resemble the insertion/deletion pattern ID8 [25]. The ID8 pattern seen in cells expressing Top2α-K743N differs from that seen in ID8 tumors that do not have Top2α-K743N. The cancer cells carrying K743N show an ID8 pattern as well as ID17, and the deletions are associated with transcription when Top2 is mutated. It is not surprising that the mutational signature of Top2 includes deletions, and it will be of interest to determine whether the deletions are also part of the inherent property of human Top2 proteins.

As noted above, it has been suggested that Top2β is involved in oncogenic translocations induced by Top2 targeting drugs. Top2β is differentially regulated compared to Top2α, and in mammalian cells, it is the only type II enzyme expressed in non-dividing cells [10]. Therefore, one hypothesis for the unique roles of Top2β is simply because all cells express this enzyme while only proliferating cells express Top2α. Alternately, Top2β may have unique biochemical properties. Rosenfeld and colleagues proposed that Top2β could induce long-lasting double-strand breaks during hormone-mediated transcription [27], and these observations have led to extensive debates concerning the precise mechanisms of Top2β action during transcription [12,30]. A question that remains unanswered is whether Top2β has intrinsic properties that enhance its likelihood of generating DNA damage.

To partly address this question, we carried out a genetic screening to assess the properties of Top2β with alterations in cleavage and religation in the presence of etoposide. Remarkably, five mutants that were analyzed in detail all had the property that the mutant proteins could not be viably expressed in *rad52^−^* cells, and that all the purified proteins exhibited elevated levels of DNA cleavage in the absence of inhibitors [96]. At present, we do not yet know whether the hyper cleavage Top2β mutants will exhibit the same mutagenic properties as top2-FY,RG and top2-K720N, and those studies are currently in progress.

## 5. Hyper Cleavage Topoisomerases: Biochemical and Structural Views

There are abundant three-dimensional structures that have been determined for eukaryotic Top2 [97,98]. Briefly, eukaryotic Top2 consists of a homodimer that includes an N-terminal ATPase domain, a connecting domain called the transducer that facilitates communication between the ATPase domain and the breakage/reunion core, which includes the active site tyrosine, a C-terminal dimerization domain, and a relatively poorly conserved and disordered C-terminal domain that likely regulates enzyme activity, localization, and protein/protein interactions [4] With this well determined molecular structure, a key question is what makes a Top2 mutant exhibit elevated levels of DNA cleavage? The first two mutants that were isolated, Top2α-D48N and top2-FY,RG are distant from the active site tyrosine, and no current structure suggests that these mutations affect the transesterification reaction. Therefore, the mutations likely exert an allosteric effect on progression through the catalytic cycle [99]. Regulation of progression through the catalytic cycle is critical for preventing Top2-induced DNA damage. Mutations in Top2β that ablate the ATPase domain of the protein are hyper cleavage in vitro, and cannot be viably expressed in *rad52^−^* cells [100]. Other mutations such as Top2α-K743N and some of the Top2β alleles (such as Top2β-K600T) may have a more direct effect on the transesterification reaction, or other steps in the Top2 catalytic cycle [96].

Since hyper-cleavage Top2 mutants might perturb intramolecular communication, it was of interest to determine whether a computational analysis could predict other mutants that have similar properties. The set of Top2β hyper cleavage mutants were modeled using molecular dynamics simulations and computational network analyses, and it was postulated that many of the hyper cleavage mutations map to interfacial points between structurally coupled elements [96]. Cancer genome databases were assessed for potential DNA damaging alleles, and two of four mutants tested Top2β-V111I and Top2β-K646N exhibited elevated levels of DNA cleavage. Top2β-V111I could not be expressed in *rad52^−^* cells while Top2β-K646N could be viably expressed in *rad52^−^* cells. These results highlight the relevance of allosteric mutants that can affect the cleavage/religation equilibrium of the Top2 catalytic cycle.

## 6. Pathways for Repairing Top2 Damage: The View from Mutation Induction

A variety of pathways have been described for repairing Top2 damage (reviewed in [101,102,103]). The logic of the main repair pathways that have been postulated is shown in Figure 3. The first step is recognition of the trapped Top2 complex as a “lesion“ requiring processing. Cells should have the capability of distinguishing a Top2 enzyme carrying out a normal reaction versus one that is stalled. The molecular details of this recognition remain poorly understood. Plausible mechanisms include Top2 complexes that interfere with transcription, replication, or other processes that include enzymes tracking along DNA [104,105]. The trapped complexes are marked by post-translational modifications that include SUMO and ubiquitin. These modifications likely play roles in recruiting repair factors. One fate of the covalent Top2 complex is proteolytic digestion, either by the proteasome, or by other proteases [104,106,107]. Proteolysis cannot completely remove the Top2 protein since proteases cannot hydrolyze phosphotyrosyl linkages. One pathway relies on tyrosyl phosphodiesterase. In mammalian cells, the enzyme Tdp2 has been reported to be the primary enzyme for repairing 5′-phophotyrosyl peptides bound to DNA. Tdp1, which is very active against 3′phosphotyrosyl adducts, is also important for removing 5′-phosphotyrosyl peptides in yeast [94], and is also active in chicken DT40 cells [108]. This enzyme likely plays some role in repairing 5-phophotyrosyl peptides bound to DNA in mammalian cells [109]. 

The second pathway for removing 5′-phosphotyrosyl peptides utilizes the MRN (Mre11/Rad50/Nbs1, in yeast the third subunit is termed Xrs2 and the yeast complex is termed MRX) repair complex. This complex, along with the nuclease CtIP (SAE2 in yeast) processes Spo11, a topoisomerase-like protein that generates double-strand breaks by forming an enzyme/DNA covalent complex to initiate meiotic recombination. Studies in yeast and mammalian cells showed that the MRN complex plays a significant role in processing trapped Top2 complexes [110,111,112,113]. Finally, recent work has suggested other repair pathways including a pathway using a protein ZATT, which has been suggested to stimulate Tdp2 activity even in the absence of Top2 proteolysis [114] ZATT has also been suggested to reverse covalent complexes in association with RAD54L2 independent of Tdp2 [115,116].

After the complete removal of the Top2 complex, the resulting DNA will have single or double-strand breaks. These breaks are repaired by canonical break repair pathways including homologous recombination and NHEJ. In mammalian cells, NHEJ predominates, while in yeast homologous recombination is the most critical pathway for repairing Top2 drug-induced double-strand breaks. The repair pathways described above likely play roles in repairing damage by hyper cleavage enzymes. As noted above, hyper cleavage enzymes expressed in yeast require homologous recombination to maintain viability [24,91]. Although Top2 covalent complexes formed in the presence of etoposide can be repaired by homologous recombination, NHEJ also plays a role [117]. The repair of Top2-induced single-strand breaks has not been studied in detail.

The frequency of topoisomerase-induced mutations depends on these repair functions. For example, endonuclease-deficient alleles of Mre11 result in a large increase in 2–4 nucleotide duplications [24], indicating that there is an alternate repair pathway mediated by the Mre11 pathway that is likely error-free. One important class of events that have not been studied extensively with hyper cleavage Top2 alleles are genome rearrangements. Top2α-K743N shows evidence for elevated levels of genome rearrangements [25]. Top2 cleavage induced by anti-cancer drugs such as etoposide has been shown to lead to oncogenic translocations [33,118,119]

## 7. Consequences of Defective Topoisomerase Action: Genome Instability Leading to Cancer and Other Diseases

The oncogenic potential of interfering with Top2 has been well established by the clinical experience that etoposide and other Top2 targeting drugs can lead to secondary leukemias. It is especially telling that mitoxantrone, which is used to treat multiple sclerosis is also associated with secondary malignancies in a non-cancer setting [120]. There is less information currently available regarding the leukemogenic potential of Top1 targeting drugs such as irinotecan and topotecan, with occasional case reports suggesting a possible leukemogenic potential (see for example [121]). As noted above, there is no clear evidence at present for secondary malignancies associated with Top1 targeting agents. Interestingly, there is some evidence that Top2 catalytic inhibitors have some oncogenic potential. Patients treated with bimolane for psoriasis have been reported to be at enhanced risk for certain leukemias [122]. Bimolane is bisdioxopiperazine catalytic inhibitor of Top2 that is closely related to the better-known compounds ICRF-187 (dexrazoxane) and ICRF-193 [123]. Bimolane causes cytogenetic abnormalities, but details of its mechanisms of potential disease formation remain unexplored.

On the other hand, the results described above indicate that both Top1 and Top2 have the potential to contribute to genome instability as indicated by the mutation spectra from different insertion/deletion events. There are important unanswered questions concerning the importance of topoisomerase-induced genome instability. Most importantly, how common is it? The deconvolution of mutational spectra is still a work in progress [124]. A second consideration is whether mutations in DNA repair genes lead to topoisomerase-induced mutations. The work of Reijns and colleagues relied upon RNaAseH2 deficiency to make connections between Top1 and insertion/deletion signature ID4 [64]. Repair pathways for topoisomerase-induced damage are still to be uncovered, and this may lead to evidence that suggests that topoisomerase-induced genome instability is more important than we appreciate at present.

Genome instability may give rise to other diseases than cancer. Recent studies have identified patients with developmental delay or immunodeficiency in patients carrying mutations in Top2β [125,126]. It remains to be seen whether any of the mutant topoisomerases in these patients have the potential to generate DNA damage. It is well established that mutations in the topoisomerase repair proteins Tdp1 and Tdp2 can lead to developmental defects [127,128]. These results suggest that topoisomerases, essential workhorses in DNA metabolism, will be found to be important in additional contexts that are relevant to human health.

## Figures and Tables

**Figure 1 ijms-25-10247-f001:**
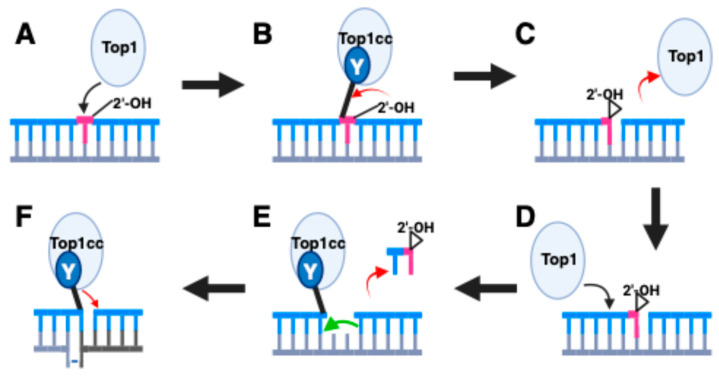
Ribonucleotide-mediated induction of deletions by Top1. (**A**) Top1-mediated deletions are initiated by cleavage specifically at a ribonucleotide (**B**) The 2′ OH of the ribose sugar attacks the phosphotyrosyl linkage. (**C**) The net reaction with the 2’ OH yields a 2′, 3′ cyclic phosphate product with the release of the Top1 protein. (**D**) A second Top1 protein cleavages upstream of the 2′, 3′ cyclic phosphate. In mammalian cells, this is frequently two nucleotides upstream. (**E**) The Top1 cleavage allows a small oligonucleotide to diffuse away. This occurs because the base pairing of 2–5 nucleotides will be relatively unstable. (**F**) Since the cleavage is in a repeat sequence, the uncleaved strand can form a short looped-out structure. This provides Top1 with an adjacent 5′ OH group to religate the break resulting in a small looped-out segment on one strand. When this heteroduplex is replicated, one of the two products will contain a deletion. Figure created with BioRender.com.

**Figure 2 ijms-25-10247-f002:**
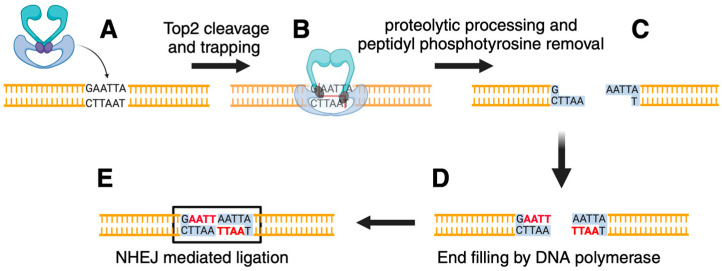
Mechanism of de novo duplication formation by trapped Top2. (**A**) Top2 interacts with DNA. (**B**) Top2 generates a cleavage complex, which can be enhanced by specific mutations or the presence of etoposide. The DNA cleavage generates a four-nucleotide overhang. (**C**) The enzyme is removed by the combined action of protease action and Tdp1 to yield a DNA with a double-strand break. The DNA break has four nucleotide overhangs, here illustrated as 5′ AATT 3′. (**D**) The overhang can be filled in by a DNA polymerase. (**E**) Upon ligation by NHEJ, the original sequence 5′ AATT 3′ becomes 5′ AATTAATT 3′. Although not shown here, a similar mechanism can explain shorter duplications [24]. Figure created with BioRender.com.

**Figure 3 ijms-25-10247-f003:**
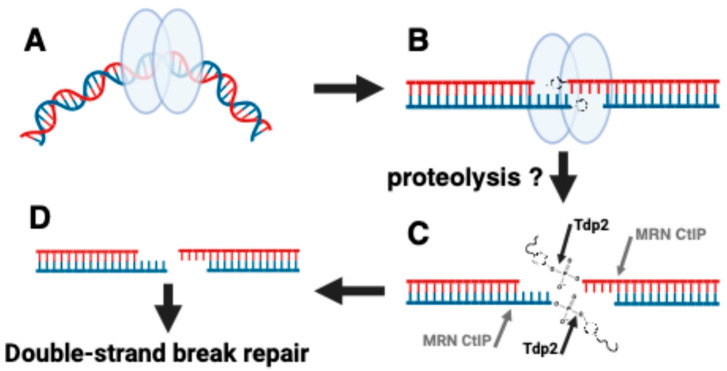
Conceptual pathways for repairing Top2 damage. The unique pathways for repairing Top2 damage vary by the mechanisms for removing Top2 protein covalently bound to DNA. (**A**) Top2 interacts with a DNA substrate, here shown as a bent DNA segment. (**B**) Initial steps can include proteolysis or potentially otherwise modify the Top2 structure to allow nucleolytic processing. (**C**) Nucleolytic processing can be carried out by proteins that can disjoin phosphotyrosyl bonds (such as Tdp2) or nucleases such as Mre11 or CtIP. (**D**) After the removal of the protein, further steps in repair can then be carried out by conventional double-strand break repair pathways. Figure created with BioRender.com.

## Data Availability

This review article contains no new data.

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
