# Peer review of "Genome Instability Induced by Topoisomerase Misfunction"

_ijms, 2024, doi:10.3390/ijms251910247_

Round 1

Reviewer 1 Report

Comments and Suggestions for Authors

In the review “Genome instability induced by topoisomerase misfunction” by Nitiss et al., genomic instability due to DNA topoisomerases during their physiological and aberrant activities is discussed. The topic is of extreme interest and has been addressed with a good level of depth. However, some parts of the review lack certain concepts, and some sentences in the various paragraphs should be rephrased. Below are the main points I have noted:

1. Introduction: The relevance of Top1 in genomic instability is not mentioned, although there is a complete paragraph about it later on. It might be better to insert some reference to this earlier;
2. Line 192: suddenly mentions camptothecin without any preliminary explanation of the mechanism. This explanation actually follows a few lines later with reference 14. The sentence should be reformulated by including some additional detail;
3. Line 106: should specify that the data were obtained in yeast;
4. Section 2. Topo1 and genome instability: The experiments on Top1 overexpression should be included. Additionally, the issue of ERCs, genome stability, and rDNA is completely ignored. Given that this is among the best examples of genomic instability associated with Top1, it should at least be briefly mentioned;
5. Lines 196-214: Although the authors only intended to briefly touch on the issue of "The detailed mechanisms of translocation induction by Top2 targeting drugs are beyond the scope of this review," what is reported is too limited and relies on rather old references (2001). It should be slightly more detailed and updated. Otherwise, it would be better to remove a consideration that is outdated and oversimplified.

Minor points:
- Line 197: a closing parenthesis is missing
- Line 366: a reference is missing within parentheses
- Line 435: these authors are not cited in the reference list

Reviewer 2 Report

Comments and Suggestions for Authors

The current manuscript reviews the less known role of Topoisomerase (mis)function in genome instability. The differences in the mode of action of typeI and typeII TopoII enzymes is presented and discussed. It also explores the importance of this molecular mechanism for cancer process. The manuscript is well written and contains a significant number of citations of relevance for the scope of this review. Below I provide with some comments and minor points that should be addressed before acceptance.  

Comments:

Concerning TopoIIa targeting small molecules, it would be of interest for the audience to introduce the distinction between TopoII poisons and inhibitors. This is well established based on the point of blockade at the Strand Passage Reaction cycle that trigger these molecules, and the maintenance – or not- of DSBs. There is much scientific literature on that.

A simple explanation of TopoII enzyme structure, highlighting its functional domains, might be of interest to better understand the biological significance of the listed mutations.

Of interest, in this review there is a especial focus on the occurrence of ins/del events linked to TopoII misfunction: e.g. ID4, ID8 signatures (insertion/deletions of 4bp or 8 bp) in repeated sequences. This is of particular interest in my opinion. Related to this, in figure 2 the targeting sequence is a 6 bp palindrome (EcoRI target sequence). The cleaving scheme of this restriction enzyme explains further insertion of the 4 pb. Repeated sequences and palindromic sequences are not the same, so I wonder if this proposed molecular mechanism for de novo duplication of repeats it is also expected upon non-palindromic repeats. This needs to be clarified by the authors.

Minor points:

Lines 17-18: sentence needs to be rewritten for clarification (add “small molecules targeting”).

Line 57: “at the interface of DNA”.

Line 366: missing reference.

Lines 443-444: sentence needs to be rewritten for clarification (mutant/altered Top2B?)

Section 8 provides few scientific data of interest, consider no to present it as a separated section within the paper.
